# External validation of Finnish diabetes risk score (FINDRISC) and Latin American FINDRISC for screening of undiagnosed dysglycemia: Analysis in a Peruvian hospital health care workers sample

Marlon Yovera-Aldana[1]*, Edward Mezones-Holguín[2,3], Rosa Agüero-Zamora[4], Lucy Damas-Casani[5], Becky Uriol-Llanos[6], Frank Espinoza-Morales[7], Percy Soto-Becerra[8,9], Ray Ticse-Aguirre[9,10]

1 Grupo de Investigación en Neurociencias, Efectividad Clínica y Salud Pública, Universidad Científica del Sur, Lima, Perú, 2 Centro de Excelencia en Investigaciones Económicas y Sociales en Salud, Universidad San Ignacio de Loyola, Lima, Perú, 3 Epi-gnosis Solutions, Piura, Peru, 4 Facultad de Medicina, Universidad Nacional Federico Villarreal, Lima, Perú, 5 Servicio de Endocrinología, Hospital María Auxiliadora, Lima, Perú, 6 Red de Eficacia Clínica y Sanitaria, Lima, Perú, 7 Centro de Tecnologías Aplicadas a la diabetes, CAVIMEDIC, Lima, Perú, 8 Instituto de Evaluación en Tecnologías en Salud e Investigación (IETSI), Lima, Perú, 9 Universidad Continental, Huancayo, Peru, 10 Escuela de Posgrado, Universidad Peruana Cayetano Heredia, Lima, Perú

* myovera@cientifica.edu.pe

## Abstract

### Aims

To evaluate the external validity of Finnish diabetes risk score (FINDRISC) and Latin American FINDRISC (LAFINDRISC) for undiagnosed dysglycemia in hospital health care workers.

### Methods

We carried out a cross-sectional study on health workers without a prior history of diabetes mellitus (DM). Undiagnosed dysglycemia (prediabetes or diabetes mellitus) was defined using fasting glucose and two-hour oral glucose tolerance test. LAFINDRISC is an adapted version of FINDRISC with different waist circumference cut-off points. We calculated the area under the receptor operational characteristic curve (AUROC) and explored the best cut-off point.

### Results

We included 549 participants in the analysis. The frequency of undiagnosed dysglycemia was 17.8%. The AUROC of LAFINDRISC and FINDRISC were 71.5% and 69.2%; p = 0.007, respectively. The optimal cut-off for undiagnosed dysglycemiaaccording to Index Youden was ≥ 11 in LAFINDRISC (Sensitivity: 78.6%; Specificity: 51.7%) and ≥12 in FINDRISC (Sensitivity: 70.4%; Specificity: 53.9%)

**Data Availability Statement:** All relevant data are within the manuscript and its Supporting Information files.

**Funding:** The author(s) received no specific funding for this work.

**Competing interests:** The authors have declared that no competing interests exist.

## Conclusion

The discriminative capacity of both questionnaires is good for the diagnosis of dysglycemia in the healthcare personnel of the María Auxiliadora hospital. The LAFINDRISC presented a small statistical difference, nontheless clinically similar, since there was no difference by age or sex. Further studies in the general population are required to validate these results.

## Introduction

Only half of the people with diabetes mellitus (DM) in the world know they have this disease. The delay in the diagnosis of diabetes mellitus affects the costs for treatment, management of macro and microvascular complications, and quality of life [1]. South America and the Caribbean have the lowest global prevalence after Africa, but it will increase by 55% in 2045 [2]. In Peru, 19.5 new cases are detected every 1000 person-years; this rate is one of the highest reported globally [3]. The systematic screening of diabetes and the application of lifestyles will prevent complications and their prices and reduce the incidence in the medium term [4].

The diagnosis of dysglycemia, DM or prediabetes requires a laboratory test. In order to further improve the performance of the screening approach, we must apply them to the population at risk [5]. In addition, the clinical practice guidelines from the US, Europe, and certain countries from Latin America (LATAM) promote DM screening in the general population as a health policy through clinical practice rules (CPR) [6–8]. There are several CPR for DM, but the FINDRISC is the most common tool used in LATAM [9], where certain countries use it through an adapted or simplified version [10].

Adaptation of CPR is necessary and highly relevant, especially when the characteristics of the population to be diagnosed are different from the participants of the original validation study [11]. Abdominal obesity in LATAM presents a different pattern than in Europe; based on this consideration, the Latinamerican Group for the Study of Metabolic Syndrome and obesity proposed a new cut-off point for waist circumference in women (90 cm) and men (94cm), which is correlated with a visceral fat area value >100 cm2 obtained by dual X-ray absorptiometry [12]. These cut-off points correlate better to insulin resistance than Adult Treatment **Panel** III cut-off points based on the body mass index. Based on these findings and the original FINDRISC, they developed the Latin American FINDRISC (LAFINDRISC), which was validated in Colombia and Venezuela in the general population using this updated criteria [13, 14].

On the other hand, health care workers show a higher risk of DM than the general population. A condition influenced by shift work, loss of the circadian rhythm of eating, mental health impairment, and sleep disturbances [15]. Prediabetes has pathophysiological alterations as diabetes, and there are microvascular complications in the early stages [16]. Therefore, our study aimed to evaluate the external validity of FINDRISC and LAFINDRISC for undiagnosed dysglycemia in health care workers at a high complexity general hospital from Peru. Our results constitute a piece of substantial primary evidence to address the DM research in a high-risk occupational health group.

## Material and methods

### Design and setting

We carried out a cross-sectional study from 20/06/2017 to 30/09/2017 in the María Auxiliadora General Hospital (MAGH), a national health facility of the Ministry of Health, located in

southern Lima's suburban area capital city of Peru. The MAGH has 1,839 workers, of which 70% are health care personnel. It has a health network that involves around one million users affiliated to Comprehensive Health Insurance (SIS from Spanish Acronym) with subsidised public health insurance.

## Population, sample, and sampling

We included adults, residence in Lima for more than six months, and a minimum working time of three months in the MAGH. We excluded subjects with DM, pregnancy, under corticosteroid therapy (at least one month in the last year), a history of antiretroviral or oncological treatment, people disabilities to walk, personnel with medical leave due to illness, vacations, or suspension from work during the selection process.

We estimated a minimum sample size in 549 participants using Epidat 4.2 (Xunta de Galicia, Santiago de Compostela, Spain) based on a prevalence of dysglycemia (diabetes mellitus and impaired fasting glucose) in Peru of 29.4% [17], expected sensitivity values of 66% and 80% for FINDRISC and LAFINDRISC, respectively, with 95% confidence level and 5% precision. In addition, we added 10% in case of refusal to participate or absence from work. The selection process was through random sampling from a list of 1839 employees.

## Dysglycemia

The diagnosis of dysglycemia included prediabetes or diabetes. Prediabetes included impaired fasting glucose: fasting glycemia between 100 mg/dL (5.6 mmol/L) and 125 mg/dL (6.9 mmol/L or impaired glucose tolerance: two hour blood glucose after a 75 g load between 140 mg/dL (7.8 mmol/L) to 199 mg/dL (11.0 mmol/L). Newly diabetes mellitus was diagnosed by fasting blood glucose $\geq$126 mg/dl (7.0 mmol/L) or blood glucose two hours after 75 g loading $\geq$ 200 mg/dL (11.1 mmol/L) [6].

## FINDRISC and LAFINDRISC

Both questionnaires present eight items: age, body mass index, abdominal circumference, personal history of physical activity, frequency of consumption of fruits and vegetables, history of antihypertensive medication, history of high blood glucose, and family history of diabetes. The difference between the two scores lies in the cut-off point for waist circumference to define abdominal obesity; in the LAFINDRISC, these values changed from 88 to 90 cm in women and 102 to 94 cm in men. Likewise, the modified questionnaire has only two categories, while the FINDRISC has three [10] (S1 Table).

## Procedures

During the break in the working day, we assessed the eligibility criteria and requested the signing of the informed consent. Then, we scheduled a maximum of eight people per time. Trained nursing staff administered the questionnaires, the oral glucose tolerance test and collected the blood samples. In the case of night work, OGTT was performed after 48 hours.

We used a digital weight scale SECA ® (USA), calibrated daily, with an accuracy of 0.5kg and a height rod attached to the wall using the standard measurement technique. According to WHO, we place an inelastic tape measure in the middle of the distance of the coastal ridge and the anterior superior iliac spine for the abdominal circumference [18]. We prepared the glucose load in 300 ml of water containing 75 g of glucose and 1.6 g of citric acid (1 squeezed lemon). We requested a minimum fasting time of 8 hours on the appointment day. We collected the basal venous samples 2 hours after glucose loading in dry tubes and centrifuged

them in the next 30 minutes [19]. We used a COBAS 6000 (c501 module) automated analyser (ROCHE, USA), according to the Center for Disease Control and Prevention [20].

### Statistical analysis

We described categorical data by frequencies and proportions. Using generalized linear models, Poisson family, logarithmic link function, and robust variance, we calculated prevalence ratios (PR) with its 95% confidence interval for each component of the FINDRISC or LAFINDRISC. In addition, we estimated the sensitivity, specificity, positive predictive value, negative predictive value, positive likelihood ratio, negative likelihood ratio, and diagnostic odds ratio for both indices. Also, we effectuated the comparison between both areas under the Receiver Operating Characteristics (ROC) curve. We used the Youden Index to identify the score with the best discriminative capacity based on estimates of specificity and sensitivity [21]. Finally, we performed a simulation with 1000 patients to calculate those correctly diagnosed considering a dysglycemia prevalence of 29.4% based on the reports from a sizeable Peruvian cohort study (PERUDIAB). We used STATA version 17.0 (Stata Corp, College Station, Texas, USA).

### Ethics

All participants signed an written informed consent based on the principles of the Declaration of Helsinki. Subjects were free to refuse to participate at any time. The Institutional Ethics Committee for Research of the Universidad Peruana Cayetano Heredia approved the study protocol, under the code CONSTANCIA 382-13-17. We kept the data confidential through codes only the principal investigator had access to the data. We communicate the results to all patients in writing. People with dysglycemia or high LAFINDRISC scores were referred to an endocrinology outpatient clinic.

## Results

From 1,835 health care workers, we randomly selected 589 subjects and 561 workers for the oral glucose tolerance test. Finally, we included 549 participants in the analysis (Fig 1).

Seventy-seven per cent of the study subjects were female; the age range ranged from 20 to 70 years, with a median of 51 years. Seventy-five per cent presented a body mass index greater than 25 kg/m2, and 65.4% showed abdominal obesity according to Latin American criteria (90 cm for women and 94 cm for men) (**Table 1**).

### Prevalence of undiagnosed dysglycemia

We found that 17.9% (95% 14.7–21.3) had undiagnosed dysglycemia, 2.6% (CI 95% 1.4–4.2) DM and 15.2% (CI 95% 12.4–18.6) prediabetes (**Table 1**). Likewise, its prevalence was significative higher in people aged 65 years or more (44.1%), BMI $\geq$ 30 (30.5%), higher value of circumference (22.8%), with hypertension (28.2%) and history of hyperglycemia (42.9%) (**Table 2**).

### Regression models

In both adjusted regression models, we found that age ($\geq$ 65), BMI ($\geq$ 30), and history of hyperglycemia increased the probability of undiagnosed dysglycemia. Hypertension medication was associated only in crude analysis Regarding waist circumference, the European and Latin American cut-off points were not associated with this outcome in adjusted models, although the LA was associated in the crude model. The FINDRISC presented a pseudo R2 of 0.1045 (p <0.001), while LAFINDRISC had a Pseudo R2 of 0.1034 (p <0.001) (**Table 2**).

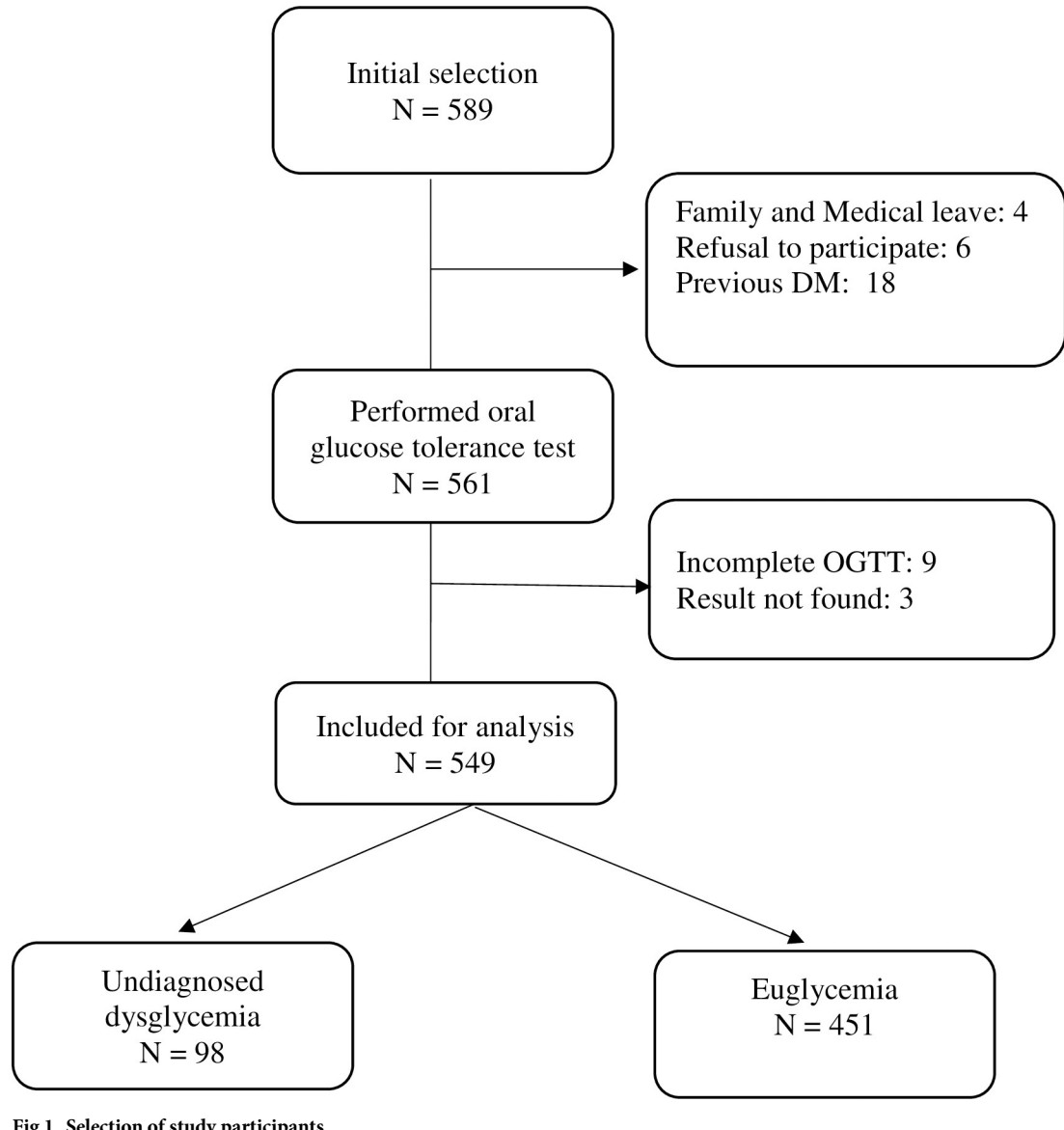

**Fig 1. Selection of study participants.**

## Comparison between scores

The discriminatory diagnostic capacity of the LAFIDNRISC was statistically greater than the FINDRISC, AUROC 71.5% (95% CI 65.8–77.2) vs 69.2% (95% CI 63.2–75.2); p = 0.007 (**Fig 2**).

When stratifying them by sex, there was similar discrimination of both questionnaires in men (61.6 vs 62.6%; p = 0.130) as in women (73.9% vs 74.8%; p = 0.338). LAFINDRISC also shows better performance for diabetes mellitus and prediabetes (**Table 3**).

In the LAFINDRISC, the score with the best Youden index was 11, showing sensitivity and specificity of 78.6% (95% CI 69.1–86.2) and 51.7% (95% CI 46.9–56.4%) respectively, and a negative likelihood ratio of 0.41 (CI95% 0.28–0.61) (**Table 4**). The supplementary material describes the complete analysis and additional comparisons (**S2 and S3 Tables**).

**Table 1. Clinical and epidemiological characteristics of healthcare workers included in the analysis.**

| | n | Percentage (CI 95%) |
|---|---|---|
| **Total** | 549 | 100 |
| **Sex** | | |
| Female | 425 | 77.4 (73.7–80.8) |
| Male | 124 | 22.6 (19.1–26.3) |
| **Type of job** | | |
| Patient care | 380 | 67.2 (65.2–73.1) |
| Administrative | 102 | 18.9 (15.4–22.1) |
| General services | 67 | 12.2 (9.6–15.2) |
| **Labour group** | | |
| Professional (university training) | 218 | 40.2 (35.6–43.9) |
| Technical–auxiliary (non-university training) | 332 | 59.7 (54.4–62.8) |
| **Age (years)** | | |
| Mean ± SD | 49.6 ± 11.2 | |
| <45 | 164 | 29.9 (26.0–33.9) |
| 45–54 | 180 | 32.8 (28.8–36.9) |
| 55–64 | 171 | 31.2 (27.3–35.2) |
| ≥ 65 | 34 | 6.2 (4.3–8.5) |
| **Body mass index (kg/m2)** | | |
| Mean ± SD | 28.2 ± 4.6 | |
| < 25 | 141 | 25.6 (22.1–29.6) |
| 25–29.9 | 255 | 46.4 (42.2–50.7) |
| ≥ 30 | 153 | 27.8 (24.2–31.8) |
| **Abdominal obesity** | | |
| Female: Mean ± SD | 93.6 ± 10.1 | |
| Male: Mean ± SD | 99.5 ± 9.5 | |
| Female ≥ 80 cm and Male ≥ 94 cm (European) | 491 | 89.4 (86.6–91.8) |
| Female ≥ 90 cm and Male ≥ 94 cm (LATAM) | 359 | 65.4 (61.2–69.3) |
| Female ≥ 88 cm and Male ≥ 102 cm (ATPIII) | 359 | 65.4 (61.2–69.3) |
| **Glycemia** | | |
| Fasting glucose in mg/dL (mmol/L) | | |
| Mean ± SD | 91.5 (5.1 mmol/l) ± 16.5 (0.9 mmol/L) | |
| <100 (< 5.6) | 460 | 83.8 (80.4–86.8) |
| 100 (5.6) to 125 (6.9) | 76 | 13.8 (11.0–17.0) |
| ≥ 126 (≥ 7) | 13 | 2.3 (1.3–4.0) |
| 2 hours post 75 g in mg/dL (mmol/L) | | |
| Mean ± SD | 98.4 (5.5 mmol/L) ± 34.5 (1.9 mmol/L) | |
| <140 (< 7.8) | 517 | 94.2 (91.9–96.0) |
| 140 (7.8) to 199 (11.0) | 25 | 4.5 (3.0–6.6) |
| ≥ 200 (≥ 11.1) | 7 | 1.3 (0.5–2.6) |
| **Glycemic disturbance** | | |
| Euglycemia | 451 | 82.2 (78.7–85.2) |
| Undiagnosed dysglycemia | 98 | 17.8 (14.7–21.3) |
| Prediabetes | 84 | 15.2 (12.4–18.6) |
| IFG | 63 | 11.4 (8.9–14.4) |
| IGT | 21 | 3.8 (2.4–5.8) |

(*Continued*)

**Table 1.** (Continued)

|  | n | Percentage (CI 95%) |
|---|---|---|
| Only IGT | 9 |  |
| IFG and IGT | 12 |  |
| Diabetes mellitus | 14 | 2.6 (1.4–4.2) |
| Only fasting glucose ≥ 126 mg/dl; ≥ 7 mmol/L | 7 |  |
| Fasting glucose ≥ 126 mg/dl; ≥ 7 mmol/L + glucose 2 h post 75g ≥ 200 mg/dl; ≥ 11.1 mmol/L | 6 |  |
| Only glucose 2 h post 75g ≥ 200 mg/dl;11.1 mmolL | 1 |  |

ATP: Adult Treatment Panel III. LATAM: Latin -american

IFG: Impaired fasting glycemia, ITG Impaired glucose tolerance, DM: Diabetes mellitus

## Simulations according to different scenarios of prevalence

We performed a simulation in 1000 patients with dysglycemia prevalence reported by the PERUDIAB (29.4%). The negative predictive value decreased from 91.7% to 85.3%, losing 6% of the test's ability to detect people without dysglycemia (**Table 5**).

## Discussion

Our research found that one out of six healthcare subjects had undiagnosed dysglycemia. One out of seven had prediabetes, and one out of fifty had diabetes mellitus. We showed a greater discriminative capacity of LAFINDRISC than FINDRISC for screening undiagnosed dysglycemia in healthcare workers. The best cut-off points for LAFINDRISC and FINDRISC were 11 and 12, respectively. The change of the cut-off point from 102 cm to 94 cm in men and from 88 to 90 cm in women better validated the results.

FINDRISC obtained an area under the ROC curve between 85 and 87% to predict drug-treated diabetes mellitus at ten years of follow-up [22]. In a captive population of northern Colombia, LAFINDRISC obtained an area under the ROC curve of 73% for undiagnosed dysglycemia [13]. LAFINDRISC received an area under the ROC curve of 68% for undiagnosed DM [23]. In both studies, there were no differences between LAFINDRISC and original FINDRISC. In our work, the area under the ROC curve was also lower than the original, with a difference of 2.3%. in favor of LAFINDRISC (71.5% vs. 69.2%). When a clinical prediction rule is validated in a population different from the original one or when a different outcome is evaluated, the discriminatory capacity tends to decrease.

We chose the Youden index to define the best score. Due to its screening purpose, it should have a higher sensitivity than specificity [24]. In our study, a score of 14 obtained the highest Youden Index with specificity greater than sensitivity. However, we chose score 11, which presented the second-best Youden Index and the requirement of having a higher sensitivity than specificity. In Colombia, a score ≥ 8 showed the highest Youden Index with a sensitivity of 78% and specificity of 50% for dysglycemia [13]. In Peru, a cut-off point of 10 of the LAFINDRISC presented a sensitivity of 70.4% and specificity of 59.1% for undiagnosed diabetes mellitus [23]. However, the original FINDRISC validation study chose the best cut-off point if it presented a negative predictive value of 99%.This criterion ensures that 1% or less of those discarded would be false negatives. If we apply this last criterion, the cut-off point would be five, and it would imply performing a second confirmatory examination on 85.1% of the population. This policy will require a higher investment and be challenging to carry out in developing economies [22].

**Table 2. Prevalence of FINDRISC items for undiagnosed dysglicemia in health workers included in the analysis.**

| | Prevalence of undiagnosed dysglicemia | | Crude Model (n = 549) | | | Adjusted Model A (n = 549) | | | Adjusted Model B (n = 549) | | |
|---|---|---|---|---|---|---|---|---|---|---|---|
| | Subtotal / Total | % (CI95%) | PR | (95% CI) | p | Pra | (95% CI) | p | Pra | (95% CI) | p |
| **Age group (years)** | | | | | | | | | | | |
| <45 | 17 / 164 | 10.4 (6.1–16.1) | 1 | (Reference) | | 1 | (Reference) | | 1 | (Reference) | |
| 45–54 | 35 / 180 | 19.4 (13.9–26.0) | 1.88 | (1.09–3.22) | 0.022 | 1.51 | (0.90–2.52) | 0.118 | 1.52 | (0.91–2.55) | 0.113 |
| 55–64 | 31 / 171 | 18.1 (12.7–24.7) | 1.75 | (1.01–3.04) | 0.047 | 1.44 | (0.82–2.51) | 0.201 | 1.40 | (0.79–2.46) | 0.25 |
| ≥ 65 | 15 / 34 | 44.1 (27.2–62.1) | 4.26 | (2.36–7.67) | **<0.001** | 2.77 | (1.50–5.12) | **0.001** | 2.72 | (1.49–4.95) | **0.001** |
| **Body mass index (kg/m2)** | | | | | | | | | | | |
| <25 | 11 /136 | 8.1 (4.1–14.0) | 1 | (Reference) | | 1 | (Reference) | | 1 | (Reference) | |
| 25–29.9 | 40 / 259 | 15.4 (11.2–20.4) | 1.91 | (1.01–3.60) | 0.046 | 1.84 | (0.99–3.42) | 0.052 | 1.38 | (0.72–2.63) | 0.33 |
| ≥ 30 | 47 / 154 | 30.5 (23.4–38.4) | 3.77 | (2.04–6.98) | **<0.001** | 3.64 | (1.89–6.99) | **<0.001** | 2.13 | (1.09–4.13) | **0.026** |
| **Waist circumference (cm)** | | | | | | | | | | | |
| M: <94 cm / F: <80 cm | 5 / 58 | 8.6 (2.9–19.0) | 1 | (Reference) | | 1 | (Reference) | | | | |
| M: 94–101.9 cm / F: 80–87.9 cm | 23 / 132 | 17.4 (11.4–25.0) | 2.02 | (0.81–5.06) | 0.133 | 1.35 | (0.55–3.35) | 0.513 | | | |
| M: ≥ 102 cm / F: ≥ 88 cm | 70 / 359 | 19.5 (15.5–24.0) | 2.26 | (0.95–5.37) | 0.064 | 0.85 | (035–2.09) | 0.726 | | | |
| **Waist circumference (cm)** | | | | | | | | | | | |
| M: <94 / F: < 90 | 16 / 190 | 8.4 (4.9–13.3) | 1 | (Reference) | | | | | 1 | (Reference) | |
| M: ≥ 94 / F: ≥ 90 | 82 / 359 | 22.8 (18.6–27.5) | 2.71 | (1.63–4.50) | **<0.001** | | | | 1.6 | (0.90–2.83) | 0.106 |
| **Regular medication hypertension** | | | | | | | | | | | |
| No | 76 / 471 | 16.1 (12.9–19.8) | 1 | (Reference) | | 1 | (Reference) | | 1 | (Reference) | |
| Yes | 22 / 78 | 28.2 (18.6–39.5) | 1.75 | (1.16–2.63) | **0.008** | 1.09 | (0.71–1.68) | 0.701 | 1.07 | (0.69–1.66) | 0.757 |
| **History of hyperglycemia** | | | | | | | | | | | |
| No | 62 / 465 | 13.3 (10.4–16.8) | 1 | (Reference) | | 1 | (Reference) | | 1 | (Reference) | |
| Yes | 36 / 84 | 42.9 (32.1–54.1) | 3.21 | (2.29–4.51) | **<0.001** | 2.42 | (1.69–3.47) | **<0.001** | 2.36 | (1.6–3.35) | **<0.001** |
| **Physical activity** | | | | | | | | | | | |
| Yes | 33 / 170 | 19.4 (13.8–26.2) | 1 | (Reference) | | 1 | (Reference) | | 1 | (Reference) | |
| No | 65 / 379 | 17.2 (13.5–21.3) | 0.88 | (0.61–1.29) | 0.521 | 0.82 | (0.58–1.17) | 0.276 | 0.8 | (0.57–1.14) | 0.217 |
| **Fruits and vegetables** | | | | | | | | | | | |
| Every day | 30 / 195 | 15.4 (10.6–21.2) | 1 | (Reference) | | 1 | (Reference) | | 1 | (Reference) | |
| Not every day | 68 / 354 | 19.2 (15.2–23.7) | 1.25 | (0.84–1.85) | 0.268 | 1.44 | (0.99–2.09) | 0.053 | 1.38 | (0.96–1.98) | 0.086 |
| **Diabetes in relatives** | | | | | | | | | | | |
| No | 44 / 275 | 16.0 (11.9–20.9) | 1 | (Reference) | | 1 | (Reference) | | 1 | (Reference) | |
| Yes, grandparents, cousins, uncle, aunt | 19 / 114 | 16.7 (10.3–24.8) | 1.04 | (0.64–1.70) | 0.871 | 1.19 | (0.73–1.95) | 0.485 | 1.1 | (0.67–1.78) | 0.714 |
| Yes, parents, siblings, son, daughter | 35 / 160 | 21.9 (15.7–29.1) | 1.37 | (0.92–2.04) | 0.125 | 1.41 | (0.97–2.04) | 0.07 | 1.3 | (0.90–1.86) | 0.158 |

PR: Prevalence rate. CI: Confidence interval 95%. Model A adjusted to components of FINDRISC. Model B adjusted to components of LAFINDRISC.M:male- F:female.

Both questionnaires presented the same performance for dysglycemia when separately analysed in men or women in our study. Nevertheless, regardless of the questionnaire used, performance in women was 12% higher than that of men. In Latin America, the area under the ROC curve of LAFINDRISC for dysglycemia in Bogotá was 76.9% in men and 77.9% in women. In Barquisimeto, the area under the ROC curve was 91.2% in men and 92.0% in women. Performance was slightly higher in women than in men in both cities [25]. In a nationwide Venezuelan study, there were no differences between FINDRISC and LAFINDRISC for dysglycemia when analyzing men and women separately [14].

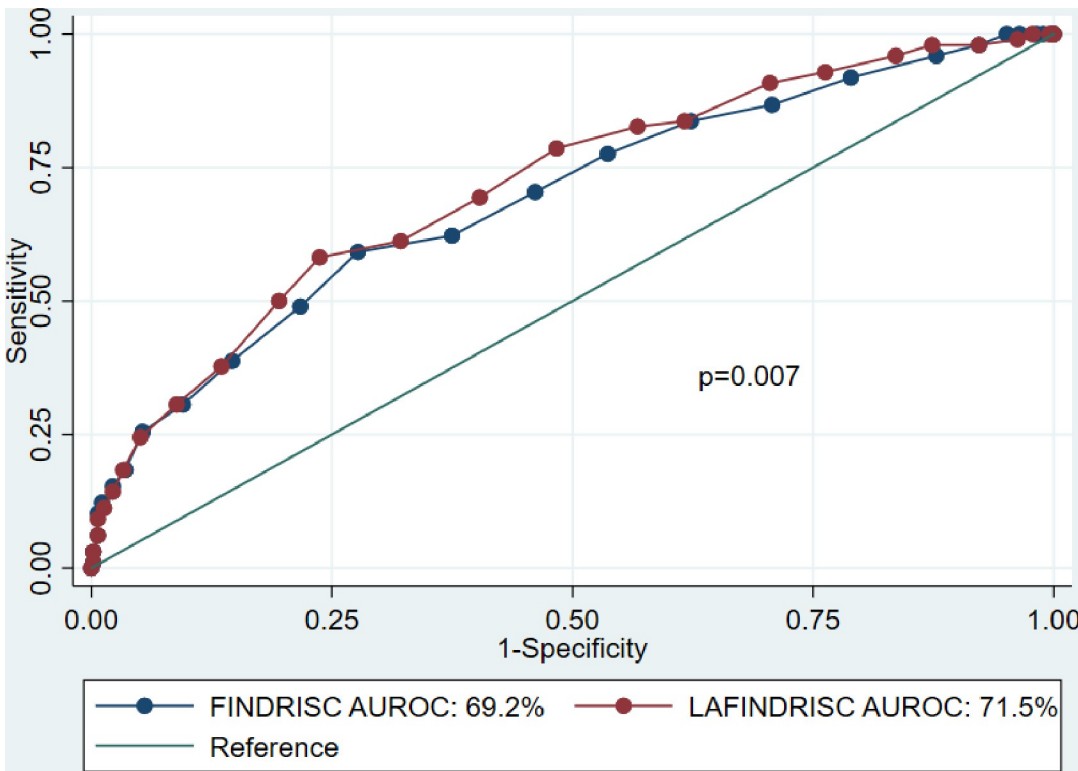

**Fig 2. Comparison of area under the ROC curves using the FINDRISC and LAFINDRISC.** AUROC: Area under receiver operating characteristic curve.

### Plausibility and explanation of results

Isolated fasting hyperglycemia implies insulin deficiency and hepatic insulin resistance but with normal muscle insulin sensitivity. This is executed by counterregulatory hormones in a context with increased lipolysis of adipose tissue and fatty esterification of liver cells that exaggerate fasting gluconeogenesis. In contrast, postprandial hyperglycemia implies a failure of secretion plus a decrease in hepatic sensitivity and moderate or high muscular resistance, preventing the internalization of glucose through GLUT4 receptors in muscle and liver. Fasting hyperglycemia could be considered an earlier failure and would predominate above all in subjects with abdominal obesity, acanthosis nigricans, skin tags or metabolic syndrome [26].

The better performance in women may be due to the high percentage of excess weight. In our study, 75% had a body mass index > 25. Likewise, the high frequency of excess weight and abdominal obesity in our healthcare workers exceeds the national average in the general population [27]. This risk is due to workgroups that perform shift work. Highlighting the nursing staff's risk of obesity and other metabolic problems represents a large percentage of healthcare workers [28].

Despite this high excess weight and abdominal obesity, our study only found 17.8% of undiagnosed dysglycemia. A difference of 11.6% concerning the national prevalence [17]. Annual occupational controls could explain this lower prevalence to detect metabolic disorders that decreased their frequency in our sample.

### Limitations and strengths

Our study has limitations. In the first place, the chosen score cannot be used in the general population since the findings would only apply to healthcare personnel of the María

**Table 3. Area under the receptor operator curves for undiagnosed dysglycemia, diabetes and prediabetes.**

| | n | FINDRISC Area under ROC curve (95% CI) | LA-FINDRISC Area under ROC curve (95% CI) | p |
|---|---|---|---|---|
| **Dysglycemia** | | | | |
| All population | 549 | 69.2% (63.2–75.2) | 71.5% (65.8–77.2) | **0.007** |
| **Sex** | | | | |
| Male | 124 | 61.6% (50.7–72.5) | 62.6% (51.9–73.3) | 0.130 |
| Female | 425 | 73.9% (66.9–80.7) | 74.8% (67.9–81.6) | 0.338 |
| **Age (years old)** | | | | |
| ≥ 45 to more | 385 | 65.3% (58.2–72.3) | 67.7% (60.9–74.4) | 0.017 |
| ≥ 55 | 205 | 68.2% (59.0–77.4) | 70.1% (61.3–78.9) | 0.103 |
| ≥ 65 | 34 | 65.4% (46.1–84.7) | 71.1 (53.7–88.4) | 0.141 |
| **Body mass index (kg/m2)** | | | | |
| ≥ 25 | 413 | 67.3% (60.8–73.9) | 69.2% (62.9–75.4) | **0.039** |
| ≥ 30 | 154 | 66.6% (57.2–75.9) | 66.2% (57.3–75.9) | 0.874 |
| **Diabetes** | | | | |
| All population | 549 | 78.6% (68.2–88.9) | 81.0% (72.0–89.9) | 0.015 |
| Male | 124 | 65.4% (39.1–91.6) | 67.6% (48.5–86.7) | 0.572 |
| Female | 425 | 80.7% (69.5–91.8) | 83.2% (73.7–92.8) | 0.005 |
| **Prediabetes** | | | | |
| All population | 535 | 67.3% (60.6–73.7) | 69.4% (63.1–75.7) | 0.020 |
| Male | 122 | 61.1% (49.9–72.4) | 62.1% (51.0–73.2) | 0.177 |
| Female | 413 | 71.7% (63.9–79.6) | 72.3% (64.4–80.1) | 0.627 |

LA-FINDRISC: latin-America Findrisc.

Auxiliadora hospital. A complimentary evaluation in a more representative population was not performed. But a simulation was performed by changing the prevalence of dysglycemia to observe the variation in performance. We do not use glycosylated hemoglobin as a

**Table 4. Discriminative characteristics of best cut-off points of FINDRISC and LAFINDRISC.**

| | Increased sensitivity | | Increased specificity | |
|---|---|---|---|---|
| | FINDRISC | LA-FINDRISC | FINDRISC | LA-FINDRISC |
| | (95% CI) | (95% CI) | (95% CI) | (95% CI) |
| **Cutt-off** | ≥ 12 | ≥ 11* | ≥ 14 | ≥ 14 |
| Youden Index | 0.24 | 0.30 | 0.31 | 0.34 |
| % correct clasification | 56.8% | 56.5% | 70.0% | 73.0% |
| Sensitivity | **70.4% (60.3% - 79.2%)** | **78.6% (69.1%– 86.2%)** | 59.2% (48.8%–69.0%) | 58.2% (47.8%–68.1%) |
| Specificity | 53.9% (49.2% - 58.6%) | 51.7% (46.9%– 56.4%) | **72.3% (67.9%–76.4%)** | **76.3% (72.1%–80.1%)** |
| Likelihood ratio positive | 1.53 (1.3–1.8) | 1.63 (1.41–1.87) | 2.14 (1.71–2.67) | 2.45 (1.94–3.10) |
| Likelihood ratio negative | 0.55 (0.4–0.75) | 0.41 (0.28–0.61) | 0.57 (0.44–0.72) | 0.55 (0.43–0.70) |
| Diagnostic odd ratio | 2.78 (1.74–4.44) | 3.92 (2.35–6.54) | 3.78 (2.41–5.93) | 4.47 (2.84–7.04) |
| Positive predictive value¶ | 24.9% (19.9% - 30.4%) | 26.1% (21.2% - 31.5%) | 31.7% (25.0%–39.0%) | 34.8% (27.5%–42.6%) |
| Negative predictive value¶ | 89.3% (85% - 92.7%) | 91.7% (87.6%-94.8%) | 89.1% (85.4%–92.1%) | 89.4% (85.8%–92.2%) |

[a] The score with the best discriminative capacity according to the Youden Index, and additionally, it had to demonstrate a higher sensitivity than specificity (See S2 and S3 Tables).

[b] Prevalence assumption of 17.9%.

Table 5. Diagnostic accuracy and implications of using a risk score*.

| Risk score | Cut-off | PPV | PPN | % sample | Dysglycemia cases detected | Subjects without dysglycemia |
|---|---|---|---|---|---|---|
| Prevalence 17.9% | | | | | | |
| FINDRISC | ≥ 12 | 24.9% | 89.3% | 50.4% | 126 | 495 |
| LAFINDRISC | ≥ 11 | 26.1% | 91.7% | 53.7% | 140 | 462 |
| Prevalence 29.4%** | | | | | | |
| FINDRISC | ≥ 12 | 38.8% | 81.4% | 53.2% | 207 | 380 |
| LAFINDRISC | ≥ 11 | 40.3% | 85.3% | 57.2% | 231 | 365 |

*All the estimates were calculated assuming 1000 individuals screened

**prevalence of dysglycemia: 20.4% from PERUDIAB

confirmatory method for dysglycemia, as we do not have methods validated by the National Gycohemoglobin Standardization Program. However, the ADA guideline recommends OGTT as a sufficient criterion for dysglycemia. Although the best discriminatory capacity of LAFINDRISC is clinically small, an instrument adapted to local characteristics is always desirable. Detailed descriptions of the participants' ethnicity were not provided as residents are commonly seen as mestizos, reflecting mixed heritage and identity reinforced by ongoing internal migration [29]. This cultural diversity has influenced Peruvian genetics, culture, and health, leading to differences in disease prevalence [30]. Although the FINDRISC is commonly used for predicting future risk of diabetes, it can also be utilized during evaluations in the present. Cross-sectional studies provide a quick snapshot of condition prevalence, identifying current at-risk individuals and validating questionnaire accuracy. They are cost-effective and time-efficient, especially useful in resource-limited settings. The data collected can also serve as a baseline for more detailed future research.

Despite these concerns, our study has important strengths, such as using a modified questionnaire with cut-off points for the Latin American obesity phenotype. In addition, subjects were randomly selected based on the sample frame of workers' payroll with minimal subject loss. OGTT was performed on all participants, regardless of the questionnaire result, avoiding selection bias.

## Implications, recommendations and future research

Quantifying the risk of diabetes or dysglycemia is a cost-effective activity recommended by the Clinical Practice Guidelines. Applying the clinical prediction rules outside the original context requires a validation process to check if the discriminative capacity is maintained. Many countries have conducted local studies and established their own thresholds [31]. Adapting these cutoff points according to ethnicity improves the accuracy in evaluating the risk of diabetes and other metabolic conditions, enabling personalized and effective interventions [32]. This consideration strengthens the clinical utility and relevance. External validations in Colombia and Peru found no differences in performance between FINDRISC and LAFINDRISC. Both studies were carried out in private insurance people and the general population, respectively [23]. Despite these results, transculturation of Clinical Prediction Rules according to the local characteristics should be the standard before applications [10].

Governments or funders will require complementary cost-effectiveness analysis and decision tree analysis for potential outcomes to apply the early diagnosis in public health [33]. It will impact the costs of screening, confirmatory diagnosis, follow-up, and treatment. As well as evaluation of potential benefits in reducing years of life gained and greater survival [34, 35].

Each country, institution, or funder chooses the cut-off point and establishes the strategy that best suits their reality. The original FINDRISC validation study determined that the best score for diabetes mellitus screening was 11. However, the Finnish Diabetes Prevention Program recommends performing OGTT from a score of 15 and initiating lifestyle changes from 7 [36]. The clinical guideline of the Colombian Ministry of Health [8] recommends performing fasting blood glucose as a confirmatory test for a score > 15 and initiating lifestyle changes if score ≥ 12. These actions are derived from decision analysis and may vary according to economic and administrative conditions.

## Conclusion

The discriminative capacity of both questionnaires is good for the diagnosis of dysglycemia in the health care personnel of the María Auxiliadora hospital. With LAFRINDRISC presenting a small statistical difference, but clinically similar since there was no difference by age or sex. Further studies in the general population are required to validate these results.

## Supporting information

**S1 Table. Scores of FINDRISC and LAFINDRISC.**
(DOCX)

**S2 Table. Performance of LAFINDRISC regarding different cut-off points.**
(DOCX)

**S3 Table. Performance of FINDRISC regarding different cut-off points.**
(DOCX)

## Author Contributions

**Conceptualization:** Marlon Yovera-Aldana, Ray Ticse-Aguirre.

**Data curation:** Marlon Yovera-Aldana, Lucy Damas-Casani.

**Formal analysis:** Marlon Yovera-Aldana, Edward Mezones-Holguín, Percy Soto-Becerra.

**Investigation:** Marlon Yovera-Aldana, Lucy Damas-Casani.

**Methodology:** Marlon Yovera-Aldana, Edward Mezones-Holguín, Lucy Damas-Casani, Percy Soto-Becerra, Ray Ticse-Aguirre.

**Project administration:** Marlon Yovera-Aldana, Lucy Damas-Casani.

**Resources:** Lucy Damas-Casani.

**Supervision:** Marlon Yovera-Aldana, Lucy Damas-Casani, Ray Ticse-Aguirre.

**Validation:** Marlon Yovera-Aldana, Edward Mezones-Holguín, Rosa Agüero-Zamora, Becky Uriol-Llanos, Frank Espinoza-Morales, Percy Soto-Becerra, Ray Ticse-Aguirre.

**Visualization:** Lucy Damas-Casani, Becky Uriol-Llanos, Frank Espinoza-Morales, Ray Ticse-Aguirre.

**Writing – original draft:** Marlon Yovera-Aldana, Edward Mezones-Holguín, Rosa Agüero-Zamora, Lucy Damas-Casani, Becky Uriol-Llanos, Frank Espinoza-Morales, Percy Soto-Becerra, Ray Ticse-Aguirre.

**Writing – review & editing:** Marlon Yovera-Aldana, Edward Mezones-Holguín, Rosa
Agüero-Zamora, Lucy Damas-Casani, Becky Uriol-Llanos, Frank Espinoza-Morales, Percy
Soto-Becerra, Ray Ticse-Aguirre.

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
