## [Decision Letter · Decision Letter 0]

25 Apr 2024

PONE-D-24-05300External validation of Finnish Diabetes Risk Score (FINDRISC) and Latin American FINDRISC for screening of undiagnosed dysglycemia: analysis in a Peruvian hospital health care workers sample.PLOS ONE

Dear Dr. Yovera-Aldana,

Thank you for submitting your manuscript to PLOS ONE. After careful consideration, we feel that it has merit but does not fully meet PLOS ONE’s publication criteria as it currently stands. Therefore, we invite you to submit a revised version of the manuscript that addresses the points raised during the review process.

We look forward to receiving your revised manuscript.

Kind regards,

Chikezie Hart Onwukwe

Academic Editor

PLOS ONE

Journal Requirements:

Reviewers' comments:

Reviewer's Responses to Questions

**Comments to the Author**

1. Is the manuscript technically sound, and do the data support the conclusions?

Reviewer #1: Yes

Reviewer #2: Yes

2. Has the statistical analysis been performed appropriately and rigorously? 

Reviewer #1: Yes

Reviewer #2: Yes

3. Have the authors made all data underlying the findings in their manuscript fully available?

Reviewer #1: Yes

Reviewer #2: Yes

4. Is the manuscript presented in an intelligible fashion and written in standard English?

Reviewer #1: Yes

Reviewer #2: Yes

5. Review Comments to the Author

Reviewer #1: Dear authors

Excellent and original work was presented. Often, the topic of prediabetes is underestimated and not given importance. In our institution, we conducted a survey on the knowledge of diabetes among doctors of other specialties than endocrinologists, and unfortunately the results were devastating. It would be nice if you stated how many respondents had a family history of early cardiovascular disease. We should definitely increase awareness of prediabetes, and get all specialties, including nurses and technicians, to pay more attention to dysglycemia. I kindly ask the authors, if the article is accepted, to MANDATORY state the glycemic values in mmol/L in the tables and text, this is very important for our colleagues from Europe who calculate in different units. And another question, have you measured HbA1c, if so, it would be good to state the results.

Thank you and congratulations once again for the excellent and original work that will open the eyes of all medical professionals and the general public and emphasize this dangerous disease.

Reviewer #2: The authors investigated questionnaires and oral glucose tolerance tests to identify dysglycemia in hospital workers in Peru. They found that the AUROC of LAFINDRISC and FINDRISC were 71.5% and 69.2%, respectively (p=0.007). They concluded that both questionnaires have good discriminative capacity for diagnosing dysglycemia in healthcare personnel.

Major criticism:

Ideally, a longitudinal study should be conducted within a specific country to assess the risk of metabolic syndrome for diabetes mellitus or cardiovascular diseases.

Minor criticism:

1. Since ethnicity (e.g., European or Indian origin) may influence risk evaluation, the authors should describe any differences among the subjects.

2. Different waist cut-off values may be necessary for diagnosing abdominal obesity in certain Asian populations. For example, in Japan, the cut-off values are 85cm for males and 90cm for females. Female has a larger cut-off value. This should be discussed.

6. PLOS authors have the option to publish the peer review history of their article (what does this mean?). If published, this will include your full peer review and any attached files.

Reviewer #1: **Yes: **Assist. prof. Visnja Kokic Males, MD, PhD

Reviewer #2: No

---

## [Author Response · Author response to Decision Letter 0]

22 May 2024

Dear reviewers, we appreciate the observations issued. We have diligently responded to each of them.

Reviewer #1:

1. I kindly ask the authors, if the article is accepted, to MANDATORY state the glycemic values in mmol/L in the tables and text, this is very important for our colleagues from Europe who calculate in different units. 

Dear reviewer, we have added the description of glucose in mmol/L

2. And another question, have you measured HbA1c, if so, it would be good to state the results.

In limitations it was already described that we did not measure glycosylated hemoglobin in the patients. Only the oral glucose tolerance test was proposed to be evaluated.

Reviewer #2: 

1. Ideally, a longitudinal study should be conducted within a specific country to assess the risk of metabolic syndrome for diabetes mellitus or cardiovascular diseases.

We have added a small comment in limitations :

Although the FINDRISC questionnaire is commonly used for predicting future risk of diabetes, it can also be utilized during evaluations in the present. Cross-sectional studies provide a quick snapshot of condition prevalence, identifying current at-risk individuals and validating questionnaire accuracy. They are cost-effective and time-efficient, especially useful in resource-limited settings. The data collected can also serve as a baseline for more detailed future research. Therefore, even without longitudinal follow-up, a cross-sectional design offers critical insights into the utility of the FINDRISC.

2. Since ethnicity (e.g., European or Indian origin) may influence risk evaluation, the authors should describe any differences among the subjects.

A description of the ethnicity of the participating subjects was not made because Lima, the capital, is a multicultural hub where inhabitants are commonly identified as mestizos, reflecting a lack of specific ethnicity. The high internal migration in Lima over the last 40 years further reinforces this mestizo identity. This rich mix has shaped the genetics, culture, and health of the Peruvian population, with notable differences in disease prevalence like diabetes.

The ethnic mixing in Peru began with the Spanish conquest in the 16th century, when European settlers established themselves in a region inhabited by advanced indigenous civilizations like the Incas. Initially, mestizaje was limited due to geography, rigid social structures, and the small number of settlers, but systems like the encomienda and the mita facilitated some mixing. Throughout the 19th and 20th centuries, new waves of immigration from Chinese, Japanese, and European populations increased ethnic diversity. In contrast, countries like Brazil experienced a more complex ethnic mix involving indigenous peoples, European settlers, and African slaves, resulting in a highly diverse population. Similarly, Mexico saw extensive mestizaje due to a larger number of Spanish settlers and a more integrated system of encomiendas. Argentina and Uruguay, with significant European immigration during the 19th and 20th centuries, saw their indigenous populations diminish and their genetic makeup become predominantly European.

We have added a small comment in limitations 

Detailed descriptions of the participants' ethnicity were not provided as residents are commonly seen as mestizos, reflecting mixed heritage and identity reinforced by ongoing internal migration. (1) This cultural diversity has influenced Peruvian genetics, culture, and health, leading to differences in disease prevalence (2).

1. Homburger JR, Moreno-Estrada A, Gignoux CR, Nelson D, Sanchez E, Ortiz-Tello P, Pons-Estel BA, Acevedo-Vasquez E, Miranda P, Langefeld CD, Gravel S, Alarcón-Riquelme ME, Bustamante CD. Genomic Insights into the Ancestry and Demographic History of South America. PLoS Genet. 2015 Dec 4;11(12):e1005602. doi: 10.1371/journal.pgen.1005602. PMID: 26636962; PMCID: PMC4670080.

2. SECLEN SANTISTEBAN Segundo. Epidemiological and genetic aspects of Diabetes mellitus in the Peruvian population. Rev Med Hered [Internet]. 1996 Oct [citado 2024 Mayo 21] ; 7( 4 ): 147-149. Disponible en: http://www.scielo.org.pe/scielo.php?script=sci_arttext&pid=S1018-130X1996000400001&lng=es.

3.. Different waist cut-off values may be necessary for diagnosing abdominal obesity in certain Asian populations. For example, in Japan, the cut-off values are 85cm for males and 90cm for females. Female has a larger cut-off value. This should be discussed.

It is essential to consider the various cutoff points of abdominal circumference when validating the FINDRISC questionnaire in different contexts and ethnic groups, taking into account the recommendations of the International Diabetes Federation (IDF). For Europeans, typical values are 94 cm for men and 80 cm for women. In the case of Asians, lower thresholds are recommended, such as 90 cm for men and 80 cm for women, while for Japanese individuals, the values are 85 cm for men and 90 cm for women. However, many countries have conducted local studies and established their own thresholds. Adapting these cutoff points according to ethnicity improves the accuracy in evaluating the risk of diabetes and other metabolic conditions, enabling personalized and effective interventions. This consideration of different cutoff points in various ethnic groups strengthens the clinical utility and relevance of the FINDRISC questionnaire.

We added a paragraph commenting this issue in Implications/Recomendatios

"Many countries have conducted local studies and established their own thresholds. (3) Adapting these cutoff points according to ethnicity improves the accuracy in evaluating the risk of diabetes and other metabolic conditions, enabling personalized and effective interventions. (4) This consideration strengthens the clinical utility and relevance"

3. Peralta Andrade K, Palacio Rojas M. Abdominal circumference cut-off point: an overview. Archivos Venezolanos de Farmacología y Terapéutica . 41: 299–306. doi:10.5281/ZENODO.6981770

4. Alberti KGMM, Eckel RH, Grundy SM, Zimmet PZ, Cleeman JI, Donato KA, et al. Harmonizing the metabolic syndrome: A joint interim statement of the international diabetes federation task force on epidemiology and prevention; National heart, lung, and blood institute; American heart association; World heart federation; International atherosclerosis society; And international association for the study of obesity. Circulation. 2009;120: 1640–1645. doi:10.1161/CIRCULATIONAHA.109.192644/FORMAT/EPUB

---

## [Decision Letter · Decision Letter 1]

18 Jun 2024

External validation of Finnish Diabetes Risk Score (FINDRISC) and Latin American FINDRISC for screening of undiagnosed dysglycemia: analysis in a Peruvian hospital health care workers sample.

PONE-D-24-05300R1

Dear Dr. Yovera-Aldana,

We’re pleased to inform you that your manuscript has been judged scientifically suitable for publication and will be formally accepted for publication once it meets all outstanding technical requirements.

Kind regards,

Chikezie Hart Onwukwe

Academic Editor

PLOS ONE

Additional Editor Comments (optional):

Reviewers' comments:

Reviewer's Responses to Questions

**Comments to the Author**

1. If the authors have adequately addressed your comments raised in a previous round of review and you feel that this manuscript is now acceptable for publication, you may indicate that here to bypass the “Comments to the Author” section, enter your conflict of interest statement in the “Confidential to Editor” section, and submit your "Accept" recommendation.

Reviewer #2: All comments have been addressed

2. Is the manuscript technically sound, and do the data support the conclusions?

Reviewer #2: Yes

3. Has the statistical analysis been performed appropriately and rigorously? 

Reviewer #2: Yes

4. Have the authors made all data underlying the findings in their manuscript fully available?

Reviewer #2: Yes

5. Is the manuscript presented in an intelligible fashion and written in standard English?

Reviewer #2: Yes

6. Review Comments to the Author

Reviewer #2: The authors revised the manuscript according to the comments from reviewers properly. This paper is ready for publication.

7. PLOS authors have the option to publish the peer review history of their article (what does this mean?). If published, this will include your full peer review and any attached files.

Reviewer #2: No

---

## [Editor Report · Acceptance letter]

27 Jun 2024

PONE-D-24-05300R1 

PLOS ONE

Dear Dr. Yovera-Aldana, 

I'm pleased to inform you that your manuscript has been deemed suitable for publication in PLOS ONE. Congratulations! Your manuscript is now being handed over to our production team.

Kind regards, 

on behalf of

Dr. Chikezie Hart Onwukwe 

Academic Editor

PLOS ONE